

# Barriers to cervical cancer prevention and triage strategies: a study of knowledge, attitudes, and p16/Ki-67 dual-staining utility among high-risk women in Tuoli and Fuyun counties, Xinjiang

Hayuehashi Tali[1], Kunayimu Yeerjiang[1], Bei bei Zeng[1], Tang Rui[1], Buajier Musha[1] and Yan Wang[2]

[1] School of Public Health, Xinjiang Medical University, Urumqi, Xinjiang, China
[2] Affiliated Cancer Hospital of Xinjiang Medical University, Urumqi, Xinjiang, China

Corresponding author
Yan Wang, xjwangyan2012@163.com

## ABSTRACT

**Objective**. To investigate cervical cancer screening knowledge, attitudes, and practices among high-risk women in remote western China, and to identify socioeconomic and systemic barriers influencing screening participation. Additionally, to evaluate the comparative effectiveness of p16 staining versus p16/Ki-67 dual-staining immunocytochemistry in triaging women with cytological abnormalities or HPV-positive results, aiming to reduce unnecessary colposcopy referrals in resource-limited settings.

**Methods**. This cross-sectional study enrolled 260 women (aged 20–65 years) with cytological abnormalities or HPV-positive results from two remote counties in Xinjiang Province (January–December 2023). Participants completed structured questionnaires assessing cervical cancer knowledge, screening attitudes, and healthcare access. Cervical specimens collected *via* liquid-based cytology underwent parallel testing: conventional cytology, p16 staining, and p16/Ki-67 dual-staining, with all analyses performed by blinded pathologists.

**Results**. Among 260 high-risk women in Xinjiang, cervical cancer awareness (67.31%, 95% CI [61.50–72.90]) and screening rates (56.15%, 95% CI [50.23–62.17]) remained suboptimal. Multivariable analyses revealed significant disparities: college-educated women had 7.58-fold higher odds of awareness (95% CI [2.32–24.75]) compared to those with primary education, while public servants showed the strongest employment-based association (aOR = 11.23, 95% CI [2.64–47.83]). Mediation analysis demonstrated that health awareness fully mediated the effect of education (128.8% mediation) and nearly fully mediated the effect of employment (93.8%). Notably, 93.98% (95% CI [90.85–96.27]) expressed willingness to rescreen, and 82.95% (95% CI [78.33–86.84]) supported HPV vaccination. Biomarker analysis showed that p16/Ki-67 dual-staining positivity increased progressively with lesion severity ($P < 0.001$).

**Conclusion**. This study reveals suboptimal cervical cancer knowledge and screening rates among women in Xinjiang, with socioeconomic disparities—particularly in education and employment—primarily mediated through health awareness. The findings support integrated interventions, including physician-led education, digital health communication for media-dependent populations, simplified visual materials for less-educated women, and active linkage to national screening programs for
unemployed populations. High rescreening willingness and parental acceptance of HPV vaccination indicate strong potential for intervention. p16 staining and p16/Ki-67 dual-staining show promise for triage in resource-limited settings. These findings highlight the need for tailored strategies to enhance cervical cancer prevention in western China, with further research needed to address current limitations.

# INTRODUCTION

As of 2023, global cervical cancer prevalence continues its upward trajectory (*Siegel et al., 2023*). In China, despite nationwide screening initiatives since 2009 (*National Health Commission of China, 2009*), stark regional disparities persist. Central and western provinces demonstrate incidence rates of 80 cases per 100,000 women, a situation exacerbated by suboptimal screening participation rates (*Siegel et al., 2023*). This disparity stems from multifactorial barriers encompassing both socioeconomic determinants (*e.g.*, impoverished living conditions, inadequate sanitation infrastructure) and systemic challenges (including required multiple clinical visits and prolonged diagnostic waiting periods). These barriers collectively result in elevated screening discontinuation rates, most notably among women with positive initial results (*Kombe Kombe et al., 2021*; *Wang et al., 2015*).

Cervical cancer screening globally, implemented similarly in both China and the United States (*Li et al., 2023*; *Wentzensen et al., 2017*), incorporates cytology, HPV testing, colposcopy, and histopathological examination. However, admitting every patient with a positive HPV test could lead to overloading the triage system (*Cuschieri et al., 2018*; *Liao et al., 2018*), high rates of colposcopy referral, and overtreatment driven by anxiety (*Gilbert et al., 2022*; *Wipperman, Neil & Williams, 2018*). Current mitigation strategies, including 12-month repeat testing protocols and genotype-specific triage algorithms (*Wentzensen et al., 2016*), have proven insufficient to establish an optimal screening pathway. This deficiency highlights the pressing need for reliable triage biomarkers. Recent advances identify p16 (*Silva et al., 2017*) and p16/Ki-67 dual-staining protocols (*Ouh et al., 2024*; *Yu et al., 2019*) as particularly promising solutions. These molecular markers provide direct visualization of HPV-mediated cellular transformation, enabling more accurate risk stratification while reducing unnecessary interventions (*Clarke et al., 2019*; *Rossi et al., 2017*).

The primary objective of this study was to conduct a cross-sectional investigation in Tuoli and Fuyun counties, remote regions of western China with limited access to cytopathologists, by combining questionnaire surveys with parallel analyses of three diagnostic modalities: conventional cytological evaluation, p16 staining, and p16/Ki-67 dual-staining. Through questionnaires, we aimed to optimize the screening experience for high-risk women identified during initial screening and explore factors influencing their follow-up adherence. Simultaneously, the laboratory-based component rigorously compared the clinical utility of p16 staining alone *versus* p16/Ki-67 dual-staining as

triage tools, with the dual goals of reducing patient anxiety and alleviating colposcopists' workload. Ultimately, this research sought to identify actionable strategies to improve recall effectiveness among high-risk populations, enhance women's understanding of screening-related knowledge, and optimize overall screening outcomes.

## METHODS

### Study design and participants

A cross-sectional study was conducted on Chinese women aged 20-65 years with cytological abnormalities or HPV positivity who received treatment or were hospitalized in the Department of Obstetrics and Gynecology at local hospitals in Tuoli and Fuyun counties, Xinjiang Province, during the period from January to December 2023.

Eligible participants were local residents with an intact cervix, sexual exposure history, non-pregnant status, no history of cervical cancer or hysterectomy, and no prior records of cone biopsy, cryotherapy, laser ablation, loop electrosurgical excision procedure (LEEP), or pelvic radiation. All participants demonstrated comprehension of study procedures and provided voluntary written informed consent.

The screening protocol integrated a standardized gynecological examination with concurrent questionnaire administration (Fig. 1). During the examination, cervical specimens were collected using liquid-based cytology (ThinPrep®) and systematically allocated for three parallel diagnostic approaches: (1) conventional cytological evaluation, (2) p16 immunocytochemical staining, and (3) p16/Ki-67 dual-staining. From each patient sample, three independent slides were prepared under identical fixation conditions to ensure analytical consistency across all testing modalities.

### Data collection

Five researchers from Xinjiang Medical University and all research assistants completed a standardized one-week training program covering questionnaire administration, interview techniques, and ethical requirements. All participants provided written informed consent after receiving detailed verbal explanations of the study objectives and procedures.

The questionnaire was developed in Chinese by a panel of researchers and medical practitioners based on a detailed examination of published literature, and has been used with established reliability and accuracy in a number of cervical cancer studies (*Zhang et al., 2021*). It comprised: (1) demographic information of participants, including marital status, level of education, employment status (classified as: 1 = unemployed; 2 = service sector workers (*e.g.*, retail and hospitality professionals such as mall salespersons, supermarket cashiers, restaurant staff); 3 = public servants (government and state employees, including military personnel, civil servants); 4 = general laborers (agricultural and factory workers such as crop farmers, factory assembly-line workers)), and income; and (2) an assessment of the participants' awareness and attitudes towards cervical cancer and its screening. The questionnaire took an average of 20–30 min to complete.

Two members of the research team entered the survey responses into Excel. To ensure data quality, all surveys were double-entered to keep the error rate below 0.5%.

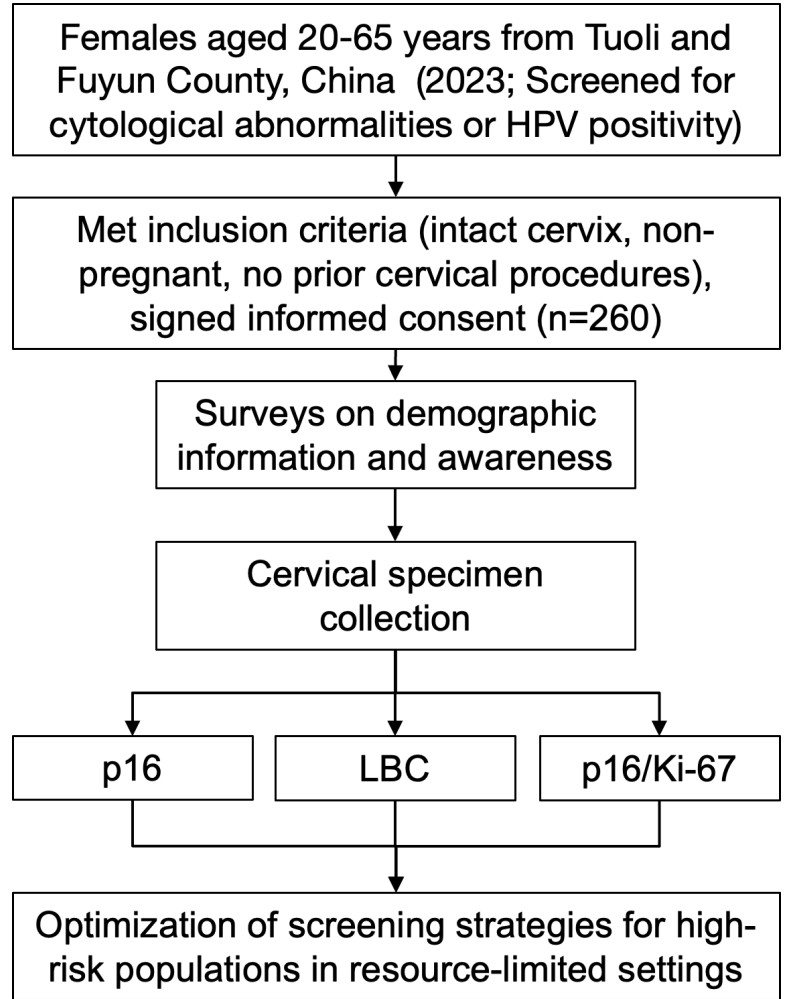

**Figure 1** Flowchart of the study design.

### p16 staining

The detection of p16 protein in cervical cytology specimens was performed using PathCIN p16 liquid-based cell preservation solution with an automated immunohistochemical staining system (Shenzhen Senying Biotech Co., Ltd., Shenzhen, China) according to the manufacturer's protocol. Positive staining was identified by the presence of brownish-yellow cytoplasmic and/or nuclear coloration in cervical epithelial cells. Staining interpretation followed the standardized scoring system established by *Wentzensen et al. (2005)*.

### p16/Ki-67 dual staining

Dual staining was conducted using the CINtec PLUS kit (CINTec® Plus Cytology Kit; Ventana Medical Systems, Inc., Oro Valley, AZ, USA) with monoclonal antibodies against p16 (clone E6H4) and Ki67 (clone 274-11 AC3). A positive result was defined as concurrent

red cytoplasmic staining (p16-positive) and brownish-yellow nuclear staining (Ki-67-positive) within the same epithelial cell on cytology slides, while a negative result included any of the following patterns: nuclei with only light blue hematoxylin counterstaining, isolated red cytoplasmic staining (p16-positive alone), or isolated brownish-yellow nuclear staining (Ki-67-positive alone). Interpretation criteria adhered to the standardized protocol described by *Celewicz et al. (2018)*.

## Liquid-Based Cytology (LBC) testing

Cervical cytology specimens were processed using ThinPrep® LBC (Hologic, Marlborough, MA, USA). Slides were independently interpreted by board-certified cytopathologists following the 2015 Bethesda Classification (*Wilbur et al., 2015*). All slides underwent blinded dual review by senior cytopathologists at Xinjiang Tumor Hospital, with discordant cases resolved by consensus. Final reports were quality-controlled by expert cytopathologists from the Cancer Hospital of the Chinese Academy of Medical Sciences, ensuring diagnostic consistency with national standards.

## Statistical analyses

The population was categorized in the statistical analysis based on participant characteristics, including screening behavior and awareness of cervical cancer. Univariate analysis of demographic information was performed using the chi-squared test and Fisher's exact test. Logistic regression models included variables associated with $P$-values < 0.05 (*Hosmer Jr, Lemeshow & Sturdivant, 2013*), with results presented as 95% confidence intervals (95% CI) and odds ratios (OR). We assessed potential multicollinearity among covariates using variance inflation factors (VIF), with all VIF values < 2.0, well below the conventional threshold of 5, indicating no substantial multicollinearity in our model. To account for multiple comparisons, we applied the Bonferroni correction. We evaluated model fit using Hosmer-Lemeshow tests (grouped by deciles) and McFadden's pseudo $R^2$. The mediation effects of educational attainment and employment status on cervical cancer screening through health awareness pathways were examined using bootstrap analysis with 5,000 resampling iterations. Fisher's exact test was performed to determine whether LBC testing was associated with p16 staining or p16/Ki-67 dual staining. Data from the study were analyzed, and radar plots were generated using STATA 18.0 software. Bar charts were drawn using Microsoft Excel software. All $P$-values <0.05 were considered statistically significant.

## Ethical consensus

This study was conducted in accordance with the Declaration of Helsinki and approved by the Ethics Committee of the Affiliated Tumor Hospital of Xinjiang Medical University (approval code: K-201802). Written informed consent was obtained from all participants prior to screening. Participants were assigned unique project identifiers, with all data and medical records maintained under strict confidentiality. Individualized written screening reports were provided, and cases with abnormal findings received prompt clinical follow-up.

## RESULTS

### Sociodemographic correlates of cervical cancer awareness and screening

This study of 260 high-risk women (HPV+/LBC abnormalities) in Xinjiang revealed 67.31% (175/260, 95% CI [61.50–72.90]) screening awareness and 56.15% (146/260, 95% CI [50.23–62.17]) participation rates, with significant sociodemographic variations (Table 1). Unmarried women (single/divorced/widowed) demonstrated substantially higher awareness than married counterparts (94.44%, 95% CI [84.55–100.00] *vs* 65.29%, 95% CI [59.26–71.34], $P = 0.009$). A pronounced education gradient was observed, with college-educated women showing both greater awareness (87.37%, 95% CI [80.63–94.17] *vs* 38.46% [23.23–53.77] in primary-educated, $P < 0.001$) and screening participation (67.37%, 95% CI [58.55–76.25] *vs* 50.00% , 95% CI [33.76–66.24], $P = 0.029$). Employment status differences were particularly striking, with public servants exhibiting the highest rates of both awareness (89.19%, 95% CI [80.55–97.85]) and screening (70.27%, 95% CI [58.16–82.44]), compared to 25.00% (95% CI [15.29–34.71]) awareness and 26.79% (95% CI [17.23–36.37]) screening among unemployed women (all $P < 0.001$). Although higher income ($\geq$100,000 RMB) initially correlated with awareness (81.82%, 95% CI [72.15–91.49] *vs* 58.95–72.73% in lower income groups; $P = 0.045$), this association became non-significant after multiplicity adjustment ($\alpha = 0.0125$).

### Factors associated with cervical cancer awareness and screening: multivariable analyses

Multivariable logistic regression analysis (Tables 2–3) revealed that college-educated women had significantly higher odds of cervical cancer screening awareness compared to those with primary education (aOR = 7.575, 95% CI [2.318–24.751]). Employment status disparities persisted after adjustment, with public servants demonstrating the strongest association (aOR = 11.230, 95% CI [2.637–47.826]), followed by service workers (aOR = 9.160, 95% CI [3.537–23.723]) and general laborers (aOR = 8.503, 95% CI [3.704–19.517]). For screening behavior, service workers (aOR = 5.671, 95% CI [2.520–12.763]), public servants (aOR = 5.118, 95% CI [1.869–14.015]), and general laborers (aOR = 3.751, 95% CI [1.817−7.745]) all showed significantly higher participation rates than unemployed women.

Both regression models demonstrated excellent calibration (awareness model: $\chi^2 = 7.010$, $df = 8$, $P = 0.535$; screening model: $\chi^2 = 3.330$, $df = 6$, $P = 0.767$). The awareness model explained a moderate proportion of variance (McFadden's pseudo $R^2 = 0.256$), considerably higher than the screening model (pseudo $R^2 = 0.081$; typical range: 0–1). This difference suggests the potential influence of additional non-sociodemographic factors on screening behavior.

### Mediating role of educational attainment and employment status on the association between awareness and screening

The bootstrap mediation analysis (5,000 resamples, adjusted for marital status and income) demonstrated complete mediation of employment status' effect on screening through health

**Table 1  Demographic correlates of cervical cancer awareness and screening behavior ($n = 260$).**

| Variables | Cervical cancer awareness | | P-value | Cervical cancer screening | | P-value |
|---|---|---|---|---|---|---|
| | Awareness | Unawareness | | Ever screened | Never screened | |
| **Marital status** | | | 0.009[**] | | | 0.351[*] |
| Married | 158 (65.29) | 84 (34.71) | | 134 (55.37) | 108 (44.63) | |
| Single/divorced/widowed | 17 (94.44) | 1 (5.56) | | 12 (66.67) | 6 (33.33) | |
| **Educational attainment** | | | <0.001[*] | | | 0.029[*] |
| Primary or lower | 10 (38.46) | 16 (61.54) | | 13 (50.00) | 13 (50.00) | |
| High school | 82 (59.85) | 55 (40.15) | | 69 (50.36) | 68 (49.64) | |
| College and higher | 83 (87.37) | 12 (12.63) | | 64 (67.37) | 31 (32.62) | |
| **Employment status** | | | <0.001[*] | | | <0.001[*] |
| Unemployed | 14 (25.00) | 42 (75.00) | | 15 (26.79) | 41 (73.21) | |
| Service sector workers | 53 (81.54) | 12 (18.46) | | 45 (69.23) | 20 (30.77) | |
| Public servants | 33 (89.19) | 4 (10.81) | | 26 (70.27) | 11 (29.73) | |
| General laborers | 73 (73.74) | 26 (26.26) | | 58 (58.59) | 41 (41.41) | |
| **Annual incomes** (RMB) | | | 0.045[*] | | | 0.744[*] |
| ≤30,000 | 24 (72.73) | 9 (27.27) | | 19 (57.58) | 14 (42.42) | |
| 30,000–60,000 | 56 (58.95) | 39 (41.05) | | 49 (51.58) | 46 (48.42) | |
| 60,000–100,000 | 55 (70.51) | 23 (29.49) | | 46 (58.97) | 32 (41.03) | |
| ≥100,000 | 36 (81.82) | 8 (18.18) | | 26 (59.09) | 18 (40.91) | |

Notes.
Data are provided as number (column percent).
1 Chinese Yuan (RMB) is approximately equal to 0.14 U.S. dollars (USD).
Participants that answered "don't know/refused to answer" were not included in the test.
[*]Chi-squared test.
[**]Fisher's exact test.

awareness (indirect effect = 0.061, 95%CI [0.029−0.093], $p < 0.001$; direct effect = 0.004, $p = 0.872$), accounting for 93.8% of the total effect (Fig. 2). A secondary reverse pathway showed employment modestly enhanced awareness through prior screening (indirect effect = 0.024, $p = 0.026$). For educational attainment, health awareness mediated 128.8% of its total effect (indirect effect = 0.103, 95%CI [0.046−0.161], $p < 0.001$), completely suppressing the direct negative association (direct effect = −0.023, $p = 0.655$). Educational attainment directly increased awareness (direct effect = 0.173, 95%CI [0.083−0.263], $p < 0.001$) with negligible mediation through screening (indirect effect = 0.029, $p = 0.164$).

## Cervical cancer screening preferences among high-risk women

According to the questionnaire results from 260 high-risk women, the majority preferred annual cervical cancer screening (128/256, 50%, 95% CI [42.16–57.84]) and expected to receive results within one week (116/258, 44.96%, 95% CI [38.34–51.64]), with written reports (89/249, 35.74%, 95% CI [30.04–41.78]) being the most favored feedback method for screening results. High compliance was demonstrated in retesting willingness, as 234 of 249 respondents (93.98%, 95% CI [90.85–96.27]) agreed to re-test if recommended by doctors. Key barriers to treatment were lack of money (82/246, 33.33%, 95% CI [27.64–39.42]) and the belief that treatment was unnecessary due to no symptoms (75/246, 30.49%, 95% CI [24.97–36.43]). Additionally, 68.99% (178/258, 95% CI [63.03–74.47]) of

**Table 2  Factors associated with cervical cancer awareness: univariate and multivariable logistic regression ($n = 260$).**

| Variables | Univariable logistic regression | | Multivariable logistic regression | |
|---|---|---|---|---|
| | Crude OR [95% CI] | *P*-value | Adjusted OR [95% CI] | *P*-value |
| **Marital status** | | | | |
| Married | Ref | | Ref | |
| Single/divorced/widowed | 9.038 [1.182–69.097] | 0.034 | 10.399 [0.898–120.441] | 0.061 |
| **Educational attainment** | | | | |
| Primary or lower | Ref | | Ref | |
| High school | 2.385 [1.009–5.642] | 0.048 | 2.384 [0.867–6.553] | 0.092 |
| College and higher | 11.067 [4.090–29.942] | <0.001 | 7.574 [2.318–24.751] | 0.001 |
| **Employment status** | | | | |
| Unemployed | Ref | | Ref | |
| Service sector workers | 13.250 [5.546–31.655] | <0.001 | 9.160 [3.537–23.723] | <0.001 |
| Public servants | 24.750 [7.446–82.263] | <0.001 | 11.230 [2.637–47.826] | 0.001 |
| General laborers | 8.423 [3.969–17.876] | <0.001 | 8.503 [3.704–19.517] | <0.001 |
| **Annual incomes** (RMB) | | | | |
| ≤30,000 | Ref | | Ref | |
| 30,000–60,000 | 0.538 [0.226–1.283] | 0.162 | 0.389 [0.134–1.126] | 0.082 |
| 60,000–100,000 | 0.897 [0.362–2.222] | 0.814 | 0.613 [0.191–1.968] | 0.411 |
| ≥100,000 | 1.688 [0.571–4.986] | 0.344 | 0.765 [0.199–2.930] | 0.696 |

Notes.

OR, odds ratio; 95% CI, 95% confidence interval; Ref, referent group.

Although annual income became nonsignificant after Bonferroni correction ($\alpha = 0.0125$), we retained it in the multivariable model based on its borderline significance in univariate analysis ($P = 0.045$) and its theoretical relevance as a sociodemographic determinant.

**Table 3  Factors associated with cervical cancer screening behavior: univariate and multivariable logistic regression ($n = 260$).**

| Variables | Univariable logistic regression | | Multivariable logistic regression | |
|---|---|---|---|---|
| | Crude OR [95% CI] | *P*-value | Adjusted OR [95% CI] | *P*-value |
| **Educational attainment** | | | | |
| Primary or lower | Ref | | Ref | |
| High school | 1.015 [0.439–2.347] | 0.973 | 0.776 [0.315–1.913] | 0.581 |
| College and higher | 2.065 [0.856–4.979] | 0.107 | 1.156 [0.430–3.107] | 0.774 |
| **Employment status** | | | | |
| Unemployed | Ref | | Ref | |
| Service sector workers | 6.150 [2.786–13.578] | <0.001 | 5.671 [2.520–12.763] | <0.001 |
| Public servants | 6.461 [2.574–16.215] | <0.001 | 5.118 [1.869–14.015] | 0.001 |
| General laborers | 3.867 [1.894–7.896] | <0.001 | 3.751 [1.817–7.745] | <0.001 |

Notes.

OR, odds ratio; 95% CI, 95% confidence interval; Ref, referent group.

Marriage and income variables not included in the Screening model (not significant due to preliminary analysis).

participants had heard of the HPV vaccine, and 82.95% (214/258, 95% CI [78.33–86.84]) intended to have their children vaccinated against HPV (Fig. 3).

The survey on knowledge sources about cervical cancer and screening showed that the primary source was physician recommendation (112/289, 38.75%, 95% CI [32.76–44.93]), followed by media (84/289, 29.07%, 95% CI [23.99–34.66]) and friend/family

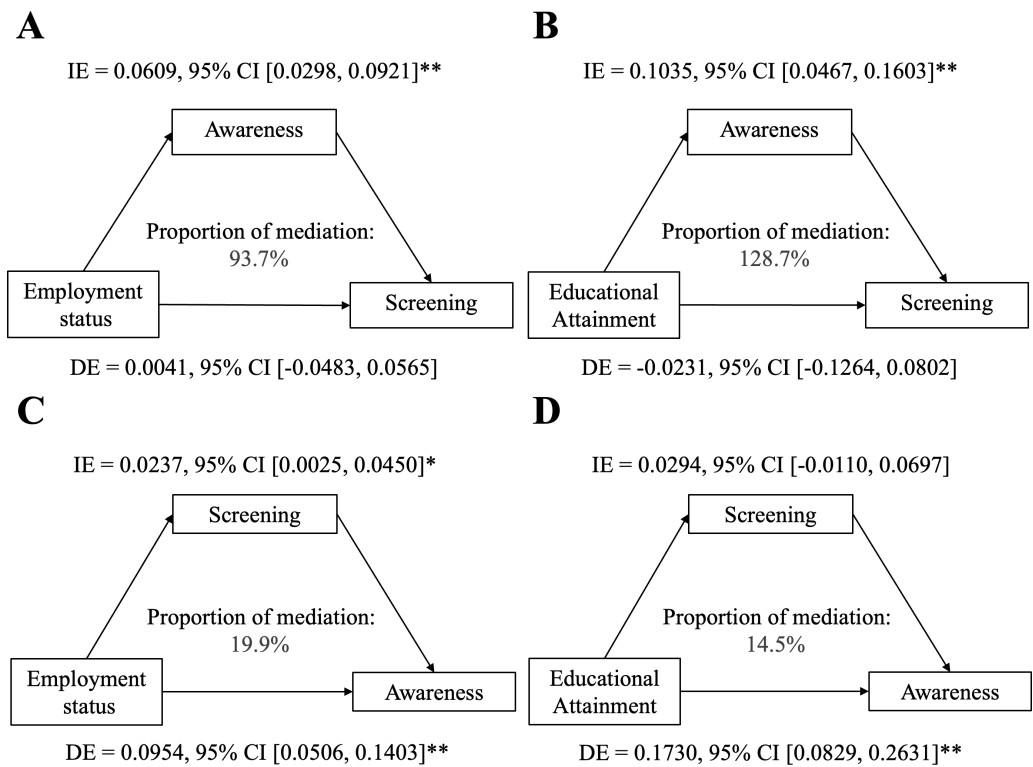

**Figure 2** (A–D) Mediation analysis of socioeconomic factors (education/employment) on screening adherence *via* health awareness, with 5,000 bootstrap resamples (*n* = 260). Covariates: marital status, income. IE = indirect effect, DE = direct effect. *P < 0.05, **P < 0.001.

recommendation (43/289, 14.88%, 95% CI [10.96–19.63]). Social welfare campaigns accounted for 12.46% (36/289, 95% CI [8.96–16.79]), while slogans/posters and other sources accounted for 2.77% (8/289, 95% CI [1.21–5.65]) and 2.08% (6/289, 95% CI [0.77–4.63]), respectively (Fig. 4).

To convince women reluctant to undergo cervical cancer screening, 37.88% (164/433, 95% CI [33.15–42.76]) of respondents prioritized increased hygiene promotion and education, followed by family/friend persuasion at 25.17% (109/433, 95% CI [21.15–29.51]). Other approaches included mandatory attendance by parent organizations (18.01%, 78/433, 95% CI [14.47–21.94]), reducing inspection frequency alongside transportation support (13.86%, 60/433, 95% CI [10.88–17.40]), and self-collecting cervical samples (3.93%, 17/433, 95% CI [2.33–6.29]), as shown in Fig. 5.

Of those unwilling to be rescreened, 29.91% (35/117, 95% CI [20.96–40.26]) cited excessive testing frequency, while 23.93% (28/117, 95% CI [15.98–33.35]) attributed it to inconvenience or distance. Other reasons included preferring county/city hospital retesting (13.68%, 16/117, 95% CI [7.69–22.47]), the screening process being too arduous (12.93%, 15/117, 95% CI [6.96–21.91]), perceived good health/non-serious results (10.26%, 12/117, 95% CI [5.20–18.17]), and time constraints (8.55%, 10/117, 95% CI [4.01–16.32]), as shown in Fig. 6.

| Awareness of cervical cancer and prevention strategies (n=260) | | |
|---|---|---|
| **Question** | **Number** | **%** |
| **How often would you be willing to undergo cervical cancer screening?** | | |
| Once per year | 128 | 50 |
| Once every three years | 42 | 16.41 |
| Once every five years | 7 | 2.73 |
| Depends on previous results | 21 | 8.2 |
| As recommended by my doctor | 55 | 21.48 |
| Unwilling | 2 | 0.78 |
| Others | 1 | 0.39 |
| **What is the maximum acceptable waiting time for receiving your screening results?** | | |
| Unsure | 12 | 4.65 |
| 1 day | 67 | 25.97 |
| Within one week | 116 | 44.96 |
| In one month | 34 | 13.18 |
| Depends on doctor's recommendation | 13 | 5.04 |
| Others | 16 | 6.2 |
| **How would you prefer to receive feedback on your screening results?** | | |
| Text message (SMS) | 25 | 10.04 |
| Telephone call | 70 | 28.11 |
| Written report | 89 | 35.74 |
| Ask the doctor yourself | 64 | 25.7 |
| Others | 1 | 0.4 |
| **Would you be willing to undergo a second test if your doctor recommended it?** | | |
| Willing | 234 | 93.98 |
| Unwilling | 5 | 2.01 |
| Uncertain | 10 | 4.02 |
| **Under what circumstances would you be unwilling to receive treatment?** | | |
| Lack of money for treatment | 82 | 33.33 |
| No symptoms, so treatment seems unnecessary | 75 | 30.49 |
| No time | 41 | 16.67 |
| Others | 48 | 19.51 |
| **Have you ever heard about the HPV/cervical cancer vaccine?** | | |
| Yes | 178 | 68.99 |
| No | 80 | 31.01 |
| **Would you want your child to receive the HPV vaccine?** | | |
| Yes | 214 | 82.95 |
| No | 44 | 17.05 |

**Figure 3** Awareness of cervical cancer and prevention strategies ($n = 260$).

## Progressive increase of p16 and p16/Ki-67 positivity correlates with the severity of cytological abnormalities detected in liquid-based cytology

Consistent with cytological findings, both p16 staining and p16/Ki-67 dual-staining exhibited a lesion-grade-dependent increase in positivity rates, escalating from 0% in normal to 100% in HSIL specimens ($P < 0.001$ for trend). This progressive increase was

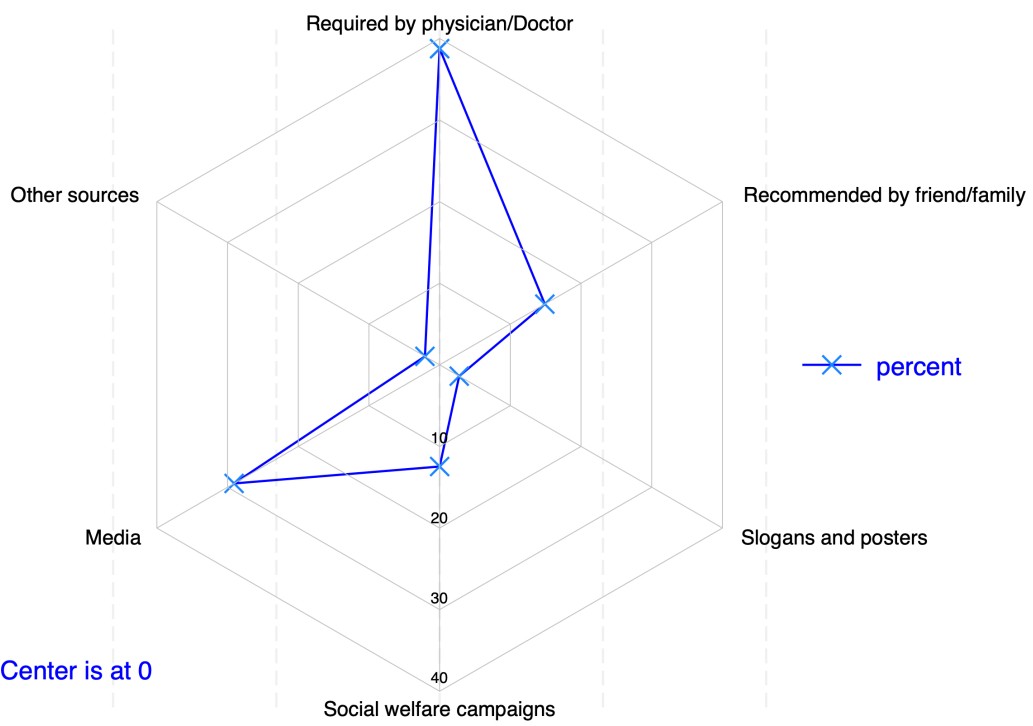

**Figure 4** Main source of information about cervical screening ($n = 289$).

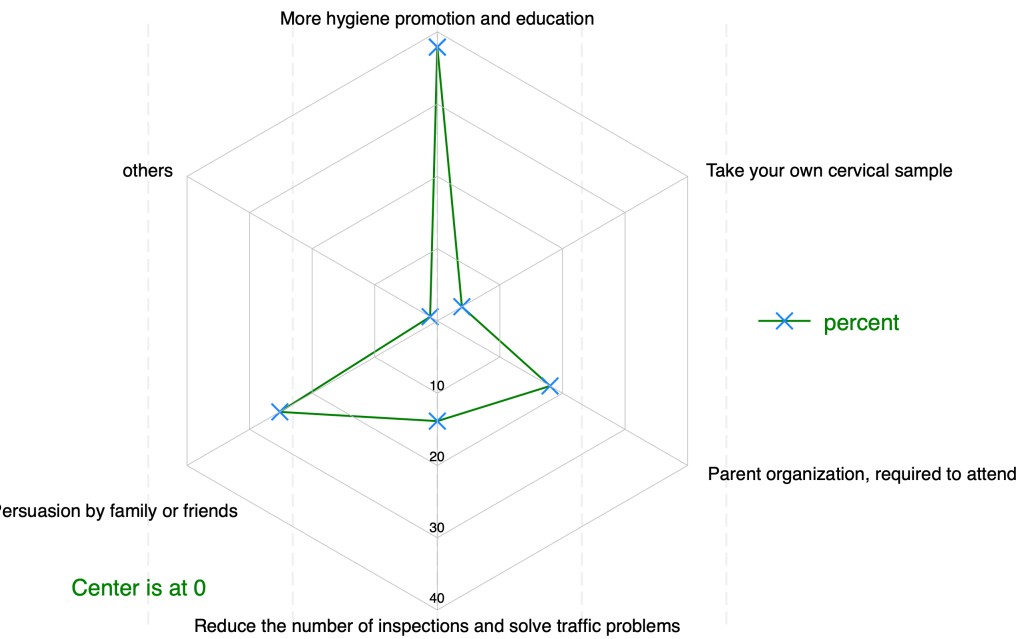

**Figure 5** Ways to persuade reluctant women to be screened for cervical cancer ($n = 433$).

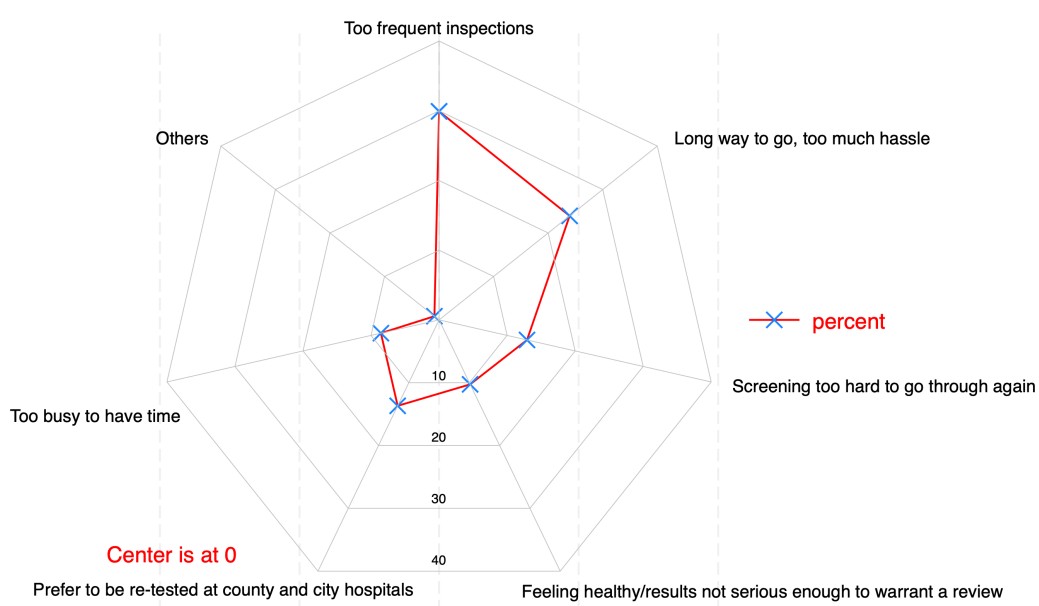

**Figure 6** Reasons given by participants for not having re-tested ($n = 117$).

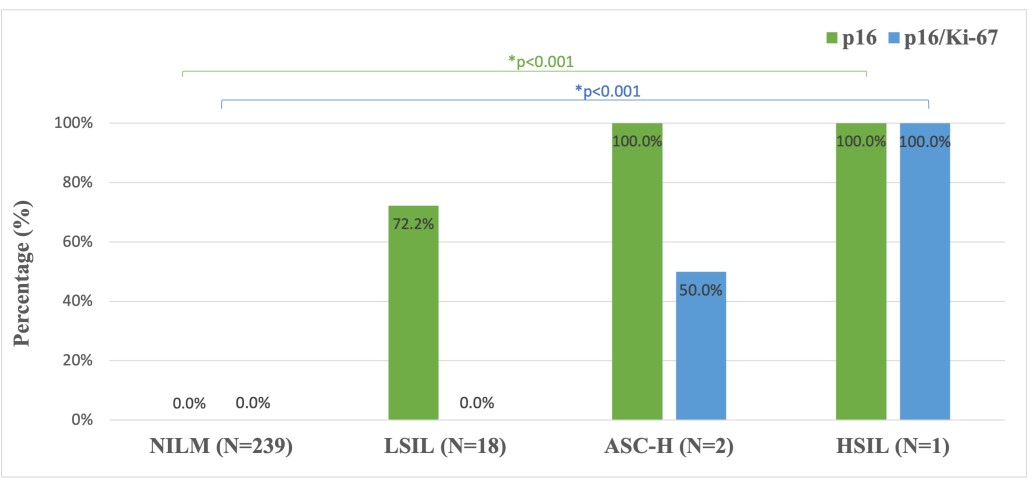

**Figure 7** p16 and p16/Ki-67 staining positivity in LBC specimens across cytological classification levels ($n = 260$). NILM, negative for intraepithe-lial lesion or malignancy; LSIL, low-grade squamous intraepithelial lesion; ASC-H, atypical squamous cells cannot exclude high-grade lesion; HSIL, high-grade squamous intraepithelial lesion; N, number;* Fisher's exact test.

observed across all diagnostic categories, with statistically significant inter-group differences (Fig. 7).

# DISCUSSION

High-risk women in Tuoli and Fuyun counties, Xinjiang, demonstrated suboptimal cervical cancer knowledge (67.31%, 175/260, 95% CI [61.50–72.90]), falling short of

China's 2025 target (>80% awareness) (*National Health Commission of China, 2021*). The rate of screening in the study population was 56.15% (146/260, 95% CI [50.23–62.17]), slightly higher than the median self-reported cervical cancer screening rate of 43.6% among women aged 30–49 years in 55 low- and middle-income countries (*Lemp et al., 2020*), but this likely reflects selection bias in our healthcare-based sampling rather than true population performance; it is still below the WHO 2030 target of 70% of women screened using a high-performance test by age 35 and again by 45 (*WHO, 2021*). These gaps highlight the need to identify socioeconomic determinants and their mediating pathways.

Multivariable analysis identified knowledge gaps as the primary modifiable barrier, consistent with prior work (*Jatho, Bikaitwoha & Mugisha, 2020*), but revealed novel mechanistic insights: first, lower educational attainment and unemployment status significantly predicted poorer awareness—college-educated women demonstrated 7.6-fold higher awareness odds (aOR = 7.575, 95% CI [2.318–24.751]) than primary-educated counterparts, while public servants showed the strongest employment-based association (aOR = 11.230 [2.637–47.826]). Second, unemployment independently reduced screening participation, with service workers (aOR = 5.671 [2.520–12.763]) and public servants (aOR = 5.118 [1.869–14.015]) exhibiting substantially higher participation than unemployed women. The mediation analysis further elucidated employment status influenced screening almost entirely through health awareness (93.8% mediation; indirect effect = 0.061 [0.029−0.093]), while education's effect was fully mediated by awareness—even reversing a direct negative association (128.8% mediation; indirect effect = 0.103 [0.046−0.161]). This pattern, aligning with but extending (*Lai, Ye & Zhao, 2025*) by quantifying the awareness-mediated pathways, suggests health education may paradoxically yield greater benefits for less-educated women.

These findings advocate for an integrated cervical cancer prevention strategy that synergistically combines screening promotion with vaccination initiatives. The data demonstrate substantial participant endorsement of health education interventions (37.88%, 95% CI [33.15–42.76]), aligning with international models (*Makadzange et al., 2022*). Physician engagement should form a cornerstone of this strategy, given their predominant role as information sources (38.7%, 95% CI [32.76–44.93]), with training programs specifically designed to counter prevalent misconceptions (*e.g.*, 30.49% believed asymptomatic women require no screening). This clinical approach requires augmentation with digital health communication platforms to effectively reach media-dependent populations (29.1%, 95% CI [23.99–34.66]), utilizing regionally accessible platforms (*e.g.*, official public accounts, short videos, and dedicated websites) to deliver accurate information (*Zhu et al., 2022*). The implementation framework should incorporate socioeconomic stratification: simplified visual aids (infographics, video tutorials) for less-educated women, and active linkage to China's national screening program for unemployed populations (*National Health Commission of China, 2021*). This tailored approach corresponds precisely with the observed education and employment gradients in screening participation. The exceptional rescreening willingness (93.98%, 95% CI [90.85–96.27]) underscores the potential for improved compliance through optimized service delivery models addressing reported barriers (23.93% cited inconvenience; 29.91%

excessive frequency). Furthermore, the strong parental vaccine acceptance (82.95%, 95% CI [78.33–86.84]) supports active promotion by clinicians to eligible populations, as evidenced in *Hendaus et al. (2021)*. Successful execution will require careful coordination through public health systems, with deliberate adaptation to local cultural norms and infrastructure realities.

Low-grade squamous intraepithelial lesions (LSIL) represent a significant proportion of abnormal cytology findings, with most cases regressing spontaneously while a minority progress to high-grade lesions or cervical cancer (*Gilbert et al., 2022*). This clinical reality necessitates reliable triage methods to identify women requiring intervention, particularly in resource-limited settings where colposcopy availability may be constrained. The FDA-approved CINtec PLUS Cytology test, which detects concurrent p16 and Ki-67 expression, has emerged as a promising solution (*Clarke, 2023*). This immunocytochemical assay identifies cell cycle dysregulation through biomarkers independent of morphological assessment (*Atkins, 2011*; *Zummeren et al., 2018*), offering several advantages: technical simplicity, cost-effectiveness, and high sensitivity for precancerous lesions (*Liao et al., 2018*; *Zhang et al., 2019*). Comparative studies demonstrate p16/Ki-67's superior performance to alternative biomarkers. It outperforms alpha-glucosidase and superoxide dismutase 2 in specificity (*Lorenzi et al., 2022*), avoids the excessive positivity rates of E6/E7 mRNA testing (*Giorgi Rossi et al., 2021*), and shows greater HSIL detection sensitivity than conventional cytology (86.7% *vs.* 68.5%) (*Shiraz et al., 2020*). When combined with morphological assessment, Ki-67 helps differentiate LSIL from normal cervical cells (*Liu, Su & Liu, 2022*), while p16/Ki-67 negativity may safely exclude the need for urgent colposcopy in HPV-positive women (*Clarke et al., 2019*; *Wentzensen et al., 2012*).

Our findings corroborate the test's diagnostic utility, showing negative correlation in normal cases and positive correlation in HSIL. The observed weak/focal positivity in LSIL and chronic inflammation aligns with existing literature (*Pirtea et al., 2019*; *Singh et al., 2012*), supporting the test's ability to risk-stratify patients. Notably, our LSIL p16 expression patterns mirror those reported by *Silva et al. (2017)*, reinforcing p16/Ki-67 positivity as a clinically significant warning for HSIL detection. For Xinjiang's resource-constrained settings lacking cytology specialists, p16/Ki-67 testing offers three key advantages: (1) technical simplicity enabling rapid HPV-positive triage within 24 h, (2) dual clinical benefits through reduced patient anxiety (by minimizing diagnostic delays) and decreased loss-to-follow-up rates (from streamlined workflows), and (3) resource optimization *via* targeted colposcopy referrals that alleviate reliance on scarce cytology specialists. This approach enhances feasibility in low-infrastructure settings while potentially yielding long-term cost savings through prevented overtreatment. However, additional validation is warranted before recommending complete replacement of histological assessment, particularly regarding long-term outcomes for test-negative women (*Giorgi Rossi et al., 2021*). Future studies should establish optimal screening intervals for p16/Ki-67 negative women and validate its performance across diverse populations.

The limitations of this research include the relatively small sample size and geographically restricted sampling within two counties, which may affect the generalizability of our findings to wider populations in Western China. We acknowledge that recruiting participants

exclusively from healthcare facilities likely introduced selection bias by overrepresenting women with better health access, potentially inflating screening awareness estimates beyond true population levels. As a cross-sectional study, our design inherently precludes causal interpretations of observed associations, such as between education level and screening awareness. Additionally, some subgroup analyses—particularly for education and employment categories—yielded estimates with wide confidence intervals, reflecting limited statistical precision due to small subgroup sample sizes. Notably, the screening behavior model explained only 8.13% of variance (McFadden's pseudo $R^2$), substantially lower than the awareness model (25.55%). This discrepancy, along with the observed suppression effect (128.8% mediation), likely stems from omitted variables (*e.g.*, health insurance coverage (*Guillaume et al., 2023*) or cultural norms) that may confound the education-screening relationship. Future studies should incorporate such constructs, particularly in population-based samples that better represent women with limited health access. While this healthcare-based sampling strategy was appropriate for our clinical research objectives, future studies would benefit from population-based sampling across diverse rural/urban settings, longitudinal designs to establish causality, and deliberate stratification by health access levels to control for selection bias. These improvements would enhance the validity and generalizability of findings while maintaining clinical relevance.

## CONCLUSIONS

This study reveals suboptimal cervical cancer knowledge (67.31%) and screening rates (56.15%) in Xinjiang, with socioeconomic disparities primarily mediated through health awareness pathways. The findings advocate for an integrated prevention strategy combining physician-led education, digital health communication for media-dependent populations, and socioeconomic-specific interventions including simplified visual materials for less-educated women and active linkage to national screening programs for unemployed populations. The exceptional rescreening willingness (93.98%) and strong parental vaccine acceptance (82.95%) underscore community readiness for interventions, while p16 staining and p16/Ki-67 dual-staining demonstrate potential for optimizing triage in resource-limited settings. Successful implementation will require coordinated public health efforts that adapt to local cultural contexts and address structural barriers, with future research needed to validate these approaches in population-based studies.

## ACKNOWLEDGEMENTS

Special thanks to the women who participated in this study and to the staff of the local hospitals.

### Funding
This study was funded by The China Medical Board (CMB) (No: 16-255) and The Science and Technology Support Project of Xinjiang (2022E02054). The funders had no role in study design, data collection and analysis, decision to publish, or preparation of the manuscript.

### Grant Disclosures
The following grant information was disclosed by the authors:
The China Medical Board (CMB): No: 16-255.
The Science and Technology Support Project of Xinjiang: 2022E02054.

### Competing Interests
The authors declare there are no competing interests.

### Author Contributions
- Hayuehashi Tali conceived and designed the experiments, performed the experiments, analyzed the data, prepared figures and/or tables, authored or reviewed drafts of the article, and approved the final draft.
- Kunayimu Yeerjiang conceived and designed the experiments, performed the experiments, prepared figures and/or tables, and approved the final draft.
- Bei bei Zeng performed the experiments, prepared figures and/or tables, and approved the final draft.
- Tang Rui performed the experiments, prepared figures and/or tables, and approved the final draft.
- Buajier Musha analyzed the data, authored or reviewed drafts of the article, and approved the final draft.
- Yan Wang conceived and designed the experiments, authored or reviewed drafts of the article, and approved the final draft.

### Human Ethics
The following information was supplied relating to ethical approvals (i.e., approving body and any reference numbers):

This study was approved by the Ethics Committee of the Affiliated Tumor Hospital of Xinjiang Medical University (project reference code K-201802) and conforms to the ethics guidelines of the Declaration of Helsinki. All the participants provided informed consent for the use of their data and samples before participation.

### Data Availability
The raw measurements are available in the Supplemental Files.

## Supplemental Information

Supplemental information for this article can be found online at http://dx.doi.org/10.7717/peerj.20100#supplemental-information.

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
