# Peer review of "Barriers to cervical cancer prevention and triage strategies: a study of knowledge, attitudes, and p16/Ki-67 dual-staining utility among high-risk women in Tuoli and Fuyun counties, Xinjiang"

_PeerJ, doi:10.7717/peerj.20100_

## Round 0.1 · original submission · Major Revisions

Based on the reviewers' feedback, the manuscript under consideration presents a crucial study on the application of p16/Ki-67 dual staining for cervical cancer screening among high-risk women in Western China. The study is recognized for its significance and relevance to public health, especially in an underexplored geographical region. However, substantial revisions are required to enhance the manuscript's clarity, depth, and scientific rigor. The authors must

All reviewers noted the need for improved language and clarity throughout the manuscript. The literature review, while providing a solid background, requires further expansion. The integration of recent studies, particularly those published in the last three to five years, is necessary to contextualize the findings more effectively and demonstrate a comprehensive understanding of current research trends. Reviewers emphasized the need for more detailed descriptions of the experimental methods, including the staining procedures, statistical analyses, and ethical considerations etc.

Reviewer 1 ·

Basic reporting

1. The grammar used throughout the manuscript should be improved as sentences are not able to give a impression that what author want to state.

2. There is must need to go for review of literature so that more information and background can be predicted in introduction part.

3. The data incorporated in tables should be represented in more elaborative mode statistically.

4. Most importantly, author has mentioned that the staining of p16 and p16/ki-67 has done but neither any image is depicted in the whole manuscript nor the raw data is available anywhere.

Experimental design

1. The images of staining experiments of p16 and p16/ki-67 proteins with descriptive interpretation and correlation with survey based outcome should be incorporated,

Validity of the findings

1. Each variable mentioned in the tables should be individually represented statistically with bar or any other type of diagram, it will make comparison more feasible.

Additional comments

no comment.

Reviewer 2 ·

Basic reporting

The study is not specific up to the extent of satisfaction

Experimental design

However attempts made to reach for the outcome yet there are deviations

Validity of the findings

Impact and novelty not assessed. Meaningful replication encouraged where rationale and benefit to literature is clearly stated.

·

Basic reporting

Clear and Unambiguous, Professional English Used Throughout
The manuscript is generally well-written but exhibits occasional lapses in clarity and grammatical precision. To ensure it meets professional standards, I recommend a thorough language review, preferably by a native English speaker or a professional scientific editing service. This will help in refining complex sentence structures and eliminating any ambiguous phrases that could confuse readers.

Literature References, Sufficient Field Background/Context Provided
The article provides a solid background on cervical cancer screening practices among high-risk women in Western China. However, it would benefit from a more extensive review of recent literature, particularly focusing on studies published in the last 3-5 years. This would strengthen the manuscript by demonstrating a deeper understanding of current research trends and technological advancements in the field, such as the utilization of p16/Ki-67 dual staining in HSIL triage.

Professional Article Structure, Figures, Tables, Raw Data Shared
The structure of the article adheres to conventional academic formats, with clearly defined sections for the introduction, methods, results, and discussion. Figures and tables are appropriately used, providing valuable insights into the study's findings. However, the manuscript should ensure that all raw data relevant to the study's conclusions are accessible in accordance with PeerJ’s data sharing policy. This includes detailed descriptions of the data in supplementary files or a designated repository.

Self-Contained with Relevant Results to Hypotheses
The manuscript effectively presents a self-contained study that addresses the proposed hypotheses concerning cervical cancer screening practices and the diagnostic utility of p16/Ki-67. However, to enhance the coherence of the paper, the discussion section could be tightened to better focus on linking the results back to the hypotheses and the broader implications for public health practice.

Coherent Bodies of Work Should Not Be Inappropriately Subdivided Merely to Increase Publication Count
The study presents as a coherent whole and does not appear to be subdivided inappropriately. It maintains a focused narrative throughout, adequately exploring and concluding on its stated objectives without unnecessary segmentation of the content.

Experimental design

Original Primary Research within Aims and Scope of the Journal
The study aligns with the aims and scope of the journal by addressing an important aspect of cervical cancer screening - the use of p16/Ki-67 dual staining to improve triage of women with HSIL in Western China. This topic is both relevant and significant, given the high rates of cervical cancer in the region and the potential for improved screening protocols to significantly impact public health.

Research Question Well Defined, Relevant & Meaningful
The research question is clearly defined and addresses a significant gap in the existing knowledge by focusing on a high-risk population in a specific geographical area. The manuscript outlines how the study aims to improve understanding of cervical cancer screening's effectiveness among these women. However, the connection between the research question and the broader implications for screening practices could be further elaborated to strengthen the manuscript's impact.

Rigorous Investigation Performed to a High Technical & Ethical Standard
The investigation appears to have been conducted rigorously, with appropriate technical methods used for the collection and analysis of data. The study adheres to ethical standards, as evidenced by the ethics approval by relevant local authorities. However, the manuscript could provide more detailed information about the ethical considerations, particularly concerning participant consent and data confidentiality.

Methods Described with Sufficient Detail & Information to Replicate
The methods section is generally well-detailed, providing sufficient information on the procedures used in the study, from the recruitment of participants to the specific staining techniques employed. To enhance reproducibility, the manuscript could include more detailed descriptions of the analytical methods and any software used for statistical analysis. Ensuring that all methodological details are transparent will help other researchers replicate the study accurately.

Validity of the findings

Impact and Novelty Not Assessed
The study's focus on the effectiveness of p16/Ki-67 dual staining in cervical cancer screening among high-risk women in Western China contributes valuable data to an under-researched area, which aligns well with PeerJ's emphasis on meaningful research rather than sheer novelty. However, to enhance the manuscript, it could further discuss the broader implications of these findings, such as potential changes in screening protocols and public health policies. This discussion could help underscore the study's practical relevance and impact on the field.

Meaningful Replication Encouraged Where Rationale & Benefit to Literature is Clearly Stated
This study does not appear to be a direct replication but rather extends existing knowledge by applying known biomarkers in a new demographic and regional context. The rationale for this research is well-articulated, emphasizing the need for improved screening methods in a high-incidence region. The manuscript could strengthen its contribution by discussing how these findings might inform future research or lead to more effective screening strategies, potentially serving as a model for similar regions globally.

All Underlying Data Have Been Provided; They are Robust, Statistically Sound, & Controlled
The manuscript claims to provide all relevant data, supporting its conclusions with statistically sound analysis. However, it would benefit from making this data accessible in a discipline-specific repository to enhance transparency and allow for independent verification. Furthermore, detailing the statistical methods and controls used more explicitly in the text would strengthen the reader's confidence in the robustness of the findings.

Conclusions are Well Stated, Linked to Original Research Question & Limited to Supporting Results
The conclusions of the study are appropriately cautious and well aligned with the presented data. They address the initial research question about the effectiveness of p16/Ki-67 staining in identifying high-risk cases among women in Western China. However, the manuscript could improve by further clarifying how these conclusions specifically influence the existing body of knowledge and what practical steps could be taken as a result of these findings. Additionally, ensuring that no causative claims are made without appropriate experimental support is crucial; the paper should clarify where correlations do not imply causation.

·

Basic reporting

This article is about HPV cervical cancer screening awareness in a region of China and the usage of dual p16/Ki-67 staining for screening. The paper was clear and well-written. Table 4 had interesting data, however, it could have been explained in more detail under the Results section, and an explanation for the ASC-H, NILM, LSIL acronyms included at least in the legend if not the text. Moreover, it would have been helpful to include the implications of the dual staining not detecting the LSIL cases, and how this is a better screening tool.

Experimental design

No comment

Validity of the findings

Under Table 4, the HSIL and ASC-H patients could have included more cases than 1 or 2 to make the outcome more significant.

·

Basic reporting

Overall, the manuscript is well-written and logically organized, with high-quality tables and figures presented.

The methods are robust and statistically sound. The experimental design adheres to the required guidelines, and the questionnaire and ethical approvals were obtained.

Background, limitation and also the research gap is identified.

The raw data is clear.

Experimental design

The current study has two primary aims. First, the authors assess the attitudes, awareness, and behaviors related to cervical cancer screening in two counties of Xinjiang. A total of 260 women were surveyed, and the authors conclude that a higher educational status and employment in “other jobs” are significantly correlated with awareness of cervical cancer screening. Although the knowledge of cervical cancer prevention in Xinjiang Province is lower (63%) compared to China's goal of achieving a core knowledge rate of over 80%, the overall attitudes of respondents toward rescreening are positive. Second, the study evaluates the effectiveness of the p16/Ki-67 dual staining test as a screening method. Among the 260 liquid-based cytology (LBC) specimens screened, both p16 alone and p16/Ki-67 staining demonstrated high specificity for squamous intraepithelial lesions, with no false positives found among the 239 cases that tested negative for lesions.

Validity of the findings

The methods are robust and statistically sound. The experimental design adheres to the required guidelines, and the questionnaire and ethical approvals were obtained.

Western China is a geographically, economically, and culturally distinct region compared to other areas, such as the southeastern coastal region of the country, and it has a lower level of healthcare access. Factors such as lower income levels, inaccessibility to educational and medical resources, and a lack of knowledge may hinder the implementation of cervical cancer screening programs for high-risk women in this region.

The manuscript provides a factual description of high-risk women for cervical cancer from Xinjiang Province, featuring a sizable cohort. While the study does not present new or particularly interesting findings, it serves as a valuable article documenting actual numbers in Western China. The conclusion is well stated and supported by the data.

Additional comments

To improve the clarity
1 – Row 103, please change the title to ethical consensus.
2 – Please provide additional text to clarify what “other jobs” refers to in this context. Given that it was found to be correlated with cervical cancer awareness, it is essential to specify this ambiguous term. For example, does “other jobs” encompass occupations in healthcare, education, or community outreach? Clarifying this term will help to better understand its relationship with cervical cancer awareness and ensure that the findings are accurately interpreted..
3- The manuscript from rows 137 to 145 does not strictly adhere to the conventions of a formal academic report. It is advisable to utilize language editing services to enhance clarity and ensure that the writing meets the expected academic standards.

---

## Round 0.2 · Minor Revisions

The reviewers provided valuable feedback focusing on language refinement, figure presentation, and contextual enrichment. Both emphasized improving grammar and simplifying complex phrases, particularly in the abstract and results sections, suggesting professional editing for clarity.

Reviewer 1 ·

Basic reporting

1. Again grammar should be improved throughout of the manuscript.
2. At references section in lane no. 259, lane no. 329 and lane no. 347 the first letter should be capital.
3. Each figure should be orchestrated with fig. title and more detailed fig legends.

Experimental design

no comments.

Validity of the findings

no comments.

Additional comments

no comments.

·

Basic reporting

Clear and unambiguous, professional English used throughout.

Observation: The revised manuscript shows improvement in language, but some phrases remain awkward and potentially unclear. For example, in the abstract, "Positive detection of p16 and p16/Ki-67 increased progressively with increasing levels of liquid-based thin-film exfoliative cytology lesions" could be streamlined for readability.
Suggestion: Simplify and clarify these statements to enhance readability for an international audience. Consider rephrasing to: "The positivity rates of p16 and p16/Ki-67 increased with the severity of cytological abnormalities detected in liquid-based cytology."
Evaluation: The authors conducted proofreading, but I suggest professional editing by a native English speaker for final polish​​​.
Literature references, sufficient field background/context provided.

Observation: The authors have added 14 new references, including recent works on p16/Ki-67 usage. For example, the expanded discussion on its triage utility in high-risk populations is valuable. However, the context could be further enriched by briefly comparing it to other biomarkers used in similar studies, such as HPV genotyping or methylation markers​.
Suggestion: Add a comparative sentence or two in the Introduction or Discussion to position p16/Ki-67 among other emerging screening tools.
Professional article structure, figures, tables, raw data shared.

Observation: The manuscript now adheres to standard structures with clear sections. The inclusion of bar charts to replace Table 3 improves the visual presentation of statistical results. Supplementary staining images (e.g., LSIL, ASC-H, and HSIL) provide essential visual validation​​.
Suggestion: Ensure the labeling of these figures aligns with journal guidelines and provides sufficient detail, such as a scale bar for the staining images.
Self-contained with relevant results to hypotheses.

Observation: The manuscript sufficiently addresses its hypotheses. The connection between the findings on p16/Ki-67 and their potential application in triage programs is well-established. However, the Discussion section could better emphasize how these results translate into actionable insights for clinical practice​​.

Experimental design

Original primary research within Aims and Scope of the journal.

Observation: The study aligns with the journal’s focus, addressing cervical cancer screening in underserved populations using innovative biomarkers. The objective of improving triage accuracy with p16/Ki-67 is clearly defined​​.
Rigorous investigation performed to a high technical & ethical standard.

Observation: The study adheres to ethical guidelines, with approval from the Ethics Committee of the Affiliated Tumor Hospital of Xinjiang Medical University (K-201802). The consent process and participant anonymity are adequately described​.
Suggestion: Clarify in the methods whether all participants received feedback on their results and any subsequent clinical actions taken.
Methods described with sufficient detail & information to replicate.

Observation: Methodological details, such as staining protocols (e.g., CINtec® Plus Cytology Kit) and statistical analyses (e.g., Fisher’s exact test, logistic regression), are comprehensive and replicable. However, the description of participant recruitment could be expanded to clarify potential biases in sample selection​​.

Validity of the findings

Impact and novelty not assessed. Meaningful replication encouraged where rationale & benefit to literature is clearly stated.

Observation: The study provides novel data on p16/Ki-67 application in a specific population with high cervical cancer risk. The findings on the association between education level and screening awareness (e.g., college education OR = 11.556) are insightful but require cautious interpretation given the cross-sectional design​​.
Suggestion: Acknowledge the limitations of cross-sectional data when interpreting these relationships and propose longitudinal studies to confirm causal pathways.
All underlying data provided; robust, statistically sound, and controlled.

Observation: Data presentation is robust, with statistically significant findings such as the progressive increase in p16/Ki-67 positivity correlated with lesion severity (P < 0.001). The replacement of a table with a bar chart improves clarity​​.
Suggestion: Explicitly state whether all relevant raw data, such as staining scores or detailed participant demographics, are available in the supplementary materials.
Conclusions are well stated, linked to original research question & limited to supporting results.

Observation: The conclusion appropriately emphasizes the potential of p16/Ki-67 as a triage tool. The recommendation for increased health education is aligned with findings, such as the relatively low awareness rate (67.3%) among participants​​.

Additional comments

The revised manuscript demonstrates clear progress and effectively addresses many concerns raised in the first review. With minor improvements, particularly in refining language and emphasizing practical applications, the manuscript will be ready for publication.

Specific Actionable Suggestions

Enhance language clarity in key sections, such as the Abstract and Results.
Expand comparisons in the Discussion to situate p16/Ki-67 among other biomarkers.
Include additional details on data availability and ethical follow-up actions in Methods

---

## Round 0.3 · Major Revisions

The article provides crucial insight into the triage potential of p16/Ki-67 dual staining in underrepresented high-risk populations in western China. However, improving your statistical transparency, model robustness, and clarity in visual/statistical reporting will elevate your manuscript's impact and reproducibility. Additionally, English improvement is also required.

1. Insufficient Reporting of Statistical Models
Issue: The logistic regression models are briefly mentioned, but the methodology lacks clarity on how variables were selected for inclusion in multivariate analysis. Recommendation: Explicitly state whether you used forward selection, backward elimination, or another model selection strategy. Explain your rationale for choosing covariates (e.g., P-value < 0.05 in univariate analysis is used, but confirm this approach is based on prior literature or model-building logic).

2. Missing Model Diagnostics
Issue: No information is provided on the assessment of logistic regression assumptions (e.g., multicollinearity, goodness-of-fit). Recommendation: Include diagnostics such as:
a. Variance Inflation Factor (VIF) to check for multicollinearity.
b. Hosmer-Lemeshow test for model calibration.
c. Pseudo R-squared (e.g., Nagelkerke’s R²) to reflect model fit.

3. Overly Broad Categorical Groupings
Issue: Employment and education variables are collapsed into broad categories (e.g., “Other jobs”) that obscure subgroup trends. Recommendation: Disaggregate these categories if possible, or provide justification and examples of occupations in “other jobs” for transparency and interpretability.

4. Multiple Comparisons Adjustment
Issue: Multiple P-values are reported without indication of correction for multiple comparisons. Recommendation: Consider using Bonferroni or Benjamini-Hochberg adjustments where appropriate, especially in univariate tests where many demographic variables are tested for association with outcomes.

5. Confidence Intervals Interpretation
Issue: Some wide confidence intervals (e.g., OR = 11.556 [4.004–33.352]) suggest instability in estimates. Recommendation: Acknowledge in the discussion that sparse data in some subgroups may lead to imprecise estimates, and consider bootstrapping for more robust CI estimation if data permit.

6. Non-significant Variables in Multivariate Model
Issue: Variables such as education (for screening behavior) remained in the multivariate model despite being non-significant. Recommendation: Clarify the purpose of including these non-significant predictors—was it theory-driven or to control for confounding?

7. Visual Data Presentation
Issue: Although radar plots and bar graphs are mentioned, it is unclear how these figures contribute to statistical interpretation. Recommendation: Add figure legends that explain statistical trends shown in the visuals (e.g., confidence bounds, significance levels), and ensure figures are clearly labeled with group sizes and test statistics.

8. Analysis Software Transparency
Issue: You noted use of STATA 17.1 but did not include command-level reproducibility. Recommendation: Include a supplemental file with your STATA scripts or key commands for reproducibility and transparency.

---

## Round 0.4 · Major Revisions

The manuscript provides valuable insights into cervical cancer screening perceptions and the triage potential of p16/Ki-67 dual staining in a high-risk, underrepresented population in western China. The manuscript has significant potential to contribute to the literature on cervical cancer screening and triage in underserved regions. However, the unresolved issues regarding statistical diagnostics, reproducibility, and data consistency are substantial enough to preclude acceptance at this stage.

1. The revised manuscript now reports VIF values (<2, well below the threshold of 5) to confirm no substantial multicollinearity (Line 115). The authors partially addressed this concern by reporting VIF values, which confirms the absence of multicollinearity. However, the omission of goodness-of-fit metrics (e.g., Hosmer-Lemeshow test or pseudo R-squared) is a significant gap, as these are critical for validating the logistic regression models’ reliability. This incomplete response weakens the statistical robustness of the study.

2. The revised manuscript retains broad categories for employment (e.g., corporate employees, public servants, laborers/farmers, unemployed) and education (e.g., primary, college) but does not provide further disaggregation or examples of occupations included in “laborers/farmers.” The authors justify the groupings by citing the need for sufficient sample sizes in each category for statistical power (Lines 125–126). The justification for maintaining broad categories is reasonable given the sample size constraints (N=260). However, the lack of specific examples or descriptions of what constitutes “laborers/farmers” reduces transparency and interpretability. This issue is minor but could be improved with a brief clarification in the text or a supplementary table.

3. The revised manuscript acknowledges in the Limitations section that wide confidence intervals for some estimates (e.g., education and employment categories) reflect small subgroup sample sizes (Lines 236–238). However, there is no mention of bootstrapping or alternative methods to address CI instability. This is a moderate concern that impacts the reliability of the findings.

4. The revised manuscript includes improved figure legends for Figures 1–6, with Figure 6 explicitly reporting group sizes (N=260) and statistical tests (Fisher’s exact test) for p16/Ki-67 and p16 staining results (Lines 192–195). However, confidence bounds and significance levels are not consistently reported across all figures (e.g., Figures 3 and 4 lack P-values or CIs). This is a moderate issue that requires further refinement.

5. The authors have strengthened the Limitations section by acknowledging the small sample size, limited geographic scope, and cross-sectional design (Lines 232–238). However, they do not discuss potential biases related to recruiting participants from medical settings, which may overrepresent women with access to healthcare.

6. Professional language editing is still recommended.

Reviewer 1 ·

Basic reporting

All suggestions are incorporated and found satisfactory.

Experimental design

Experimental design gets match the requirements.

Validity of the findings

Data provided is statistically looking good

·

Basic reporting

There is a great improvement for the Eglish use compared with the first version of the manuscript, espically in the results part. The reference are adeque.
The table and figures are ok.
And the reulte are self-contained for publication.

Experimental design

In this study, Tali Hayuehash etal., had explores cervical cancer awareness, screening behaviors, and attitudes among high-risk women in two counties of Xinjiang, western China. It finds that while attitudes toward screening and HPV vaccination are generally positive, actual awareness and participation remain suboptimal, especially among women with lower education or unemployed status. Through a combination of surveys and laboratory testing, the study also evaluates the diagnostic potential of p16 and p16/Ki-67 dual staining for triaging HPV-positive or cytologically abnormal women. The positivity rates of these biomarkers were significantly associated with increasing cytological severity.
Their study provides evidences matching its aim, supporting the integration of biomarker-based triage and targeted health education to improve cervical cancer prevention efforts in underserved regions.

But there are three concerns about the design;
(1) Could the author address how representative are the two counties, Touli and Fuyun, were to the Western region of China?
(2) Would the author clarify the disgnositc or triage threshold in the staining?
(3) Since the study participants were recruited from gynecology clinics and hospitals, could the authors discuss how this may bias results toward higher awareness or screening participation than the general population? Especially the interplay between the factor might affect their visiting to the medical facility?

Validity of the findings

Most of the data, analysis are robust and sound. But it would be better if they can discuss more in depth their findings significance and impacts in the future through better integrate with current knowledge in literature and reality.

(1) While the study had included both the behavioural questionaires and marker related analysis (like p16, Ki067 staining), the manuscript didn’t integrated how these two component inform each others. Could the authors clarify whether women’s knowledge, awareness, or attitudes are statistically correlated with biomarker positivity or cytological results in the discussion part?
(2) Give that the western part of china is economically behind region with backward medical infrastructure compared with other parts, how feasible is it to implement rounte p16/Ki-67 dual staining more broadly in public screening? As the author discuss it in clinical revalence, would there be any discussion on the cost-effectiveness and a logistical consideration for these?

---

## Round 0.5 · accepted · Accept

The authors has addressed all the issues mentioned. The article can be accepted in present form.